# A Non-Canonical Link between Non-Coding RNAs and Cardiovascular Diseases

**DOI:** 10.3390/biomedicines10020445

**Published:** 2022-02-14

**Authors:** Lucia Natarelli, Christian Weber

**Affiliations:** 1Institute for Cardiovascular Prevention (IPEK), Ludwig-Maximilians-Universität (LMU), 800336 Munich, Germany; christian.weber@med-uni.muenchen.de; 2German Center for Cardiovascular Research (DZHK), Munich Heart Alliance, 80336 Munich, Germany

**Keywords:** non-coding RNAs, cardiovascular diseases, miRNA, lncRNA, circRNA, non-canonical

## Abstract

Cardiovascular diseases (CVDs) are among the top leading causes of mortality worldwide. Besides canonical environmental and genetic changes reported so far for CVDs, non-coding RNAs (ncRNAs) have emerged as key regulators of genetic and epigenetic mechanisms involved in CVD progression. High-throughput and sequencing data revealed that almost 80% of the total genome not only encodes for canonical ncRNAs, such as micro and long ncRNAs (miRNAs and lncRNAs), but also generates novel non-canonical sub-classes of ncRNAs, such as isomiRs and miRNA- and lncRNA-like RNAs. Moreover, recent studies reveal that canonical ncRNA sequences can influence the onset and evolution of CVD through novel “non-canonical” mechanisms. However, a debate exists over the real existence of these non-canonical ncRNAs and their concrete biochemical functions, with most of the dark genome being considered as “junk RNA”. In this review, we report on the ncRNAs with a scientifically validated canonical and non-canonical biogenesis. Moreover, we report on canonical ncRNAs that play a role in CVD through non-canonical mechanisms of action.

## 1. Introduction

According to the World Health Organization (WHO), cardiovascular, respiratory, and neonatal conditions are the top global leading causes of death, causing 55% of total deaths worldwide [1]. If we exclude the coronavirus (COVID-19) pandemic disease, which accounted for almost 10% of total deaths worldwide in 2020 [2], cardiovascular diseases (CVDs), such as ischaemic heart disease, stroke, hypertension, and atherosclerosis, are among the top leading causes of morbidity and mortality, causing almost 21% of all deaths globally [3,4]. CVDs are followed by respiratory conditions, such as chronic obstructive pulmonary disease and lower respiratory infections, and by Alzheimer’s disease and *Diabetes mellitus*, responsible for almost 1.8% of deaths globally [5] (World Health Statistics 2021).

Besides canonical environmental and genetic changes reported so far for CVD, recent studies reveal the existence of new mechanisms, defined as “non-canonical”, that influence the onset and evolution of CVD. Indeed, in addition to the canonical genetic changes, non-canonical epigenetic modifications strongly influence cardiovascular disorders. Master regulators of both canonical and non-canonical modifications are the non-coding RNA (ncRNA) molecules, capable of influencing cardiovascular development and homeostasis through conventional and non-conventional mechanisms [6,7,8,9,10]. In addition, recent studies indicate that ncRNAs biogenesis and mechanisms of action are themselves regulated by non-canonical mechanisms, revealing new unconventional functions and targets influencing human biogenesis and diseases. In this review, we describe the ncRNAs showing a non-canonical biogenesis and the ncRNAs regulating CVD through non-canonical mechanisms.

## 2. Biogenesis and Development of Atherosclerosis, the Principal CVD Worldwide

Although the age-adjusted trend of CVD mortality is decreasing, at least in western European countries, cardiovascular morbidity is expected to increase in 2030 [11]. This is partly because of the increasing average age of the population, which will result in an unavoidable increase in health care costs [4,12,13]. Principal CVDs include heart failure, acute myocardial infarction, arrhythmia, and pulmonary hypertension, mainly caused by hypercholesterolemia, atherosclerosis, diabetes, and other metabolic disorders [4,14].

Atherosclerosis is usually described as a chronic inflammatory disease [15] that develops primarily in middle and large arteries, especially at branching points and curvatures. Since the flow at vessel curvatures is physiologically disturbed, endothelial cells (ECs) lining these vessel areas show a physiological alteration of their integrity and function [16,17]. Indeed, ECs lining linear arterial tracts that are exposed to laminar flow show a so-called “athero-protective” phenotype [18], characterized by a quiescent phenotype and anti-inflammatory state [17,19,20]. Conversely, ECs exposed to disturbed flow show an activated or so-called “maladapted” phenotype, characterized by a pro-inflammatory and pro-apoptotic state [21], partly balanced by a flow-mediated increase in EC proliferation [16,17,19]. Although excessive proliferation has been regarded as pro-atherogenic [19], restoring impaired EC regeneration is beneficial against atherosclerosis [22]. Hence, EC maladaptation to this physiological disturbance increases the susceptibility of branching points to a pro-inflammatory state [17,21]. During hyperlipidaemia, circulating oxidized lipoprotein influx in the sub-endothelial space is increased [23,24,25] and promotes the recruitment of monocytes through the endothelial-mediated release of pro-inflammatory molecules (e.g., adhesion molecules and chemokines) [24,26,27]. Sub-endothelial monocytes differentiate into macrophages, which locally proliferate and take up oxidized lipoproteins [20,24]. Exacerbation of lipid deposition into the plaque inhibits EC proliferation and promotes EC inflammation and apoptosis, sustaining chronic endothelial wound healing [16]. When lipoprotein scavenging from resident macrophages becomes ineffective, macrophages turn into foam cells [28]. Advanced lesion progression is characterized by an accumulation of lipoproteins, macrophage-derived foam cell apoptosis and necrosis due to defective efferocytosis [28]. Formation of cholesterol crystals and smooth muscle cell cap formation are among the last phases of plaque formation, where cap formation is initiated to prevent plaque rupture [29]. Hence, the growth of advanced lesions leads to a critical reduction of arterial lumen and blood flow, reduced oxygen supply, and rupture or erosion of the plaques, which can cause thrombosis [29].

## 3. Non-Coding RNAs

After the first discovery of the existence of small non-protein-coding RNA molecules around the 1950s, such as the 15 kb ncRNA able to regulate the inactivation of the X chromosome [30], the number of studies reporting the existence of ncRNAs exponentially increased [31,32,33,34]. We arrived at the interpretation that 80% of the genome encodes for ncRNAs [35], almost outclassing the role of protein-coding genes [33,34,35]. An interesting observation was made in 2015 by Palazzo and Lee, who discussed a singular but important controversy between the existence of ncRNAs and their concrete biochemical function [36]. Indeed, some individuals and the ENCODE consortium assumed and declared a priori that the 80% of the dark genome was functional, without the existence of a real scientific demonstration [35,36]. Apart from some criticisms linked to the cited assumption, the debate about the percentage of the dark genome being “functional” or “junk” was partly explained by Palazzo and Lee, who proposed the approach to consider uncharacterized ncRNAs as not functional, unless proven otherwise [36,37]. Moreover, they underlined the importance of considering the difference between the *types* of transcripts and the *abundance* of transcripts, which is fundamental to define putative ncRNA functions [36,37]. Hence, the real function of ncRNAs is far from being discovered.

ncRNAs can be classified according to their size in small and long non-coding RNAs [36,38,39]. The small ncRNA subclass comprises ribosomal RNAs (rRNAs), transfer RNAs (tRNAs), small nuclear and nucleolar RNAs (snRNAs, snoRNAs), signal recognition particle RNAs (e.g., 7SL), micro RNAs (miRNAs), and short enhancer RNAs (eRNAs) [36,38,39]. The rRNAs are the most abundant ncRNAs in terms of RNA mass in mammalian cells, but tRNAs represent the most abundant subclass in terms of number of molecules [40]. This underlines the concept already introduced by Palazzo and Lee about the difference between *types* and *abundance* of ncRNA transcripts when considering their biochemical function [36]. All ncRNA molecules longer than 200 nucleotides are classified as long non-coding RNAs (lncRNAs) and comprise more heterogeneous categories divided according to their genomic origin and biogenesis [36,38,39]. By general convention, lncRNAs, such as the long intergenic lncRNAs (lincRNAs), are classified as canonical ncRNAs, whereas circular RNAs (circRNAs), a sub-class of lncRNAs, are classified as non-canonical lncRNAs. circRNAs were initially classified as “junk RNA” because of their low transcription rate [32,34,35]. However, the low transcription rate of lincRNAs and circRNAs is due to their highly stable secondary structures generated after maturation (e.g., after splicing processes). Moreover, lincRNAs and circRNAs can be stored in selective subcellular compartments or expressed only under certain physiologic or pathologic conditions [36,38,39]. In addition to the *type* and *abundance* parameters used to classify ncRNAs as functional or junk RNA molecules, *biogenesis* and *mechanisms of action* are two additional parameters used to classify ncRNAs as canonical or non-canonical RNA molecules [36].

## 4. Small ncRNAs

As mentioned before, there are several methods to classify ncRNAs, but the main one is according to their size [39]. Indeed, all ncRNAs that are less than 200 bp in length are classified as small ncRNAs. They can be further grouped according to their sequence characteristics, the genomic localization of their corresponding genes, and their cross-species functional conservation. The *biogenesis* is an additional criterion to divide all small ncRNAs into seven subclasses [36,38,39], such as snRNAs, rRNAs, snoRNAs, tRNAs, miRNAs, small Cajal body-specific RNAs (scaRNAs), and intron-derived ncRNAs [36,38,39]. Among these, miRNAs and miRNA-like molecules [39], tRNAs, and snoRNAs show both a canonical and a non-canonical biogenesis, reflected also in an alternative biological function. So far, miRNAs and miRNA-like molecules are those better characterized [41], despite the existence of only a few indirect functional studies.

### 4.1. Canonical Biogenesis of miRNAs

According to the HUGO Gene Nomenclature Committee classification, miRNAs are 21–25 nt in length and exert their functional role as regulators of gene expression by targeting the mRNA transcripts of their target genes in the RNA-induced silencing complex (RISC) [41,42]. Thus, miRNAs lead to the degradation or translational repression of their target transcripts [41]. Not only protein-coding but also lncRNAs can be targets of miRNAs, indicating that miRNA plasticity is likely to affect a very large spectrum of diverse biological processes, even at the epigenetic level [9,43,44].

The biogenesis of almost 40% of miRNAs occurs within 5 min [45,46]. Genes that encode for miRNAs can be independent genes, or frequently located within introns of host genes, despite the fact that expression of the miRNA and the host gene can be independently coordinated [47]. Most of the canonical miRNA genes are transcribed by RNA polymerase II, which generates an unprocessed primary sequence named pri-miRNA [39,42,46]. Following the formation of a hairpin structure, the nuclear pri-miRNA is cleaved into a 70 nt precursor, named pre-miRNA, by the RNase III enzyme Drosha in complex with Dgcr8 to protect the precursor from degradation [39,42,46]. The cleavage of pri-miRNA by Drosha is determined by the distance between the hairpin loop and the single-stranded RNA basal segments of the pri-miRNA, recognized by Drosha for the proper cleavage [41,42,48]. The hairpin structure shuttles from the nucleus to the cytoplasm through Exportin-5 and is further processed into a mature miRNA duplex of approximately 22 nt by the endoribonuclease (RNase II) enzyme Dicer. The Dicer-mediated cleavage of the pre-miRNA hairpin loop is favoured by other RNA-binding proteins, such as TRBP, that in complex with Dicer stabilize the miRNA precursor and favour the cleavage [48,49]. Although the entire mechanism is unknown, one of the miRNA duplex strands has an increased affinity energy for Argonaute 2 (AGO2) protein in the RISC [41,42,49,50,51]. Hence, one strand of the miRNA duplex is usually degraded, whereas the second (the guide strand) persists in the RISC and interacts with the 3´-end of target RNAs [41,42,49,52] (Figure 1).

Although the current nomenclature used for the miRNA duplexes is “miRNA” and “miRNA-star” to instantly distinguish their functional context, this nomenclature is based on the miRNAs’ static characteristics [51,52]. Indeed, Cloonan et al., along with miRBase curators, underline the importance of considering that dominant strands can switch across different tissues, indicating that the expression of conventional 5p and 3p arms is not representative of a certain miRNA, especially if the same miRNA develops through non-conventional Drosha/Dicer-mediated alternative cleavages [42,53] (see Section 4.2.4).

miRNAs bind to RNAs through 2–7 nt at the 5´end of their sequence, termed “seed sequence”, to induce the degradation or the translational inhibition of the RNA targets [41,51,52] (Figure 1). Since RNA transcripts can share the same target site for different miRNAs, and an miRNA can modulate the expression of several RNAs [41,54], miRNAs emerged as master regulators and interplayers between cardiovascular and metabolic diseases [9].

### 4.2. Non-Canonical Biogenesis of Small RNAs

miRNAs produced through alternative biogenesis pathways are classified as non-canonical miRNAs. They functionally and structurally resemble canonical miRNAs. Examples of non-canonical miRNAs are the snoRNA-, short hairpin RNA (shRNA)-, tRNA-derived miRNAs, miRtrons, and the Dicer-independent miRNA subtypes [10,55] (Figure 1). In addition, all canonical miRNAs subjected to an enzymatic or biochemical sequence modification during the maturation process are classified as non-canonical miRNAs. Members of this subclass are the isomiRs [53,56] and the miRNA epitranscriptomes [57] (Figure 1).

#### 4.2.1. snoRNA-, shRNA-, and tRNA-Derived miRNAs

snoRNAs are a class of ncRNAs of 60–300 nt in length that localize primarily in the nucleolus of cells where they can modify rRNAs, snRNAs, and tRNAs. SnoRNAs can process rRNA precursors through their H/ACA box, directing the pseudouridylation and methylation of selective rRNA sites [10,58,59]. Several lines of evidence indicate that some snoRNAs can be precursors of non-canonical miRNAs, generated through a biological mechanism that seems to share similarities with the biogenesis of canonical miRNAs, such as the involvement of Dicer, Dgcr8, and AGO enzymes. The so-called snoRNA-derived miRNAs rely on Dicer and Dgcr8 for their stabilization and on Dgcr8 for their degradation [59]. Similar to miRNAs, snoRNA-derived miRNAs are 21 nt long and can bind to AGO enzymes, such as AGO1 and AGO4, to suppress mRNA targets [59]. In contrast, two snoRNA-derived miRNAs, named ACA45 and GlsR17, show a biogenesis depending not on Dicer but on Drosha/Dgcr8 [10,59]. These snoRNA-derived miRNAs are characterized by a unique precursor structure containing two pre-miRNA-like hairpins linked by a hinge. Like miRNAs, these unique structures seem to confer them the ability to post-transcriptionally silence mRNA transcripts [59] (Figure 1).

shRNA-derived miRNAs derive from non-miRtronic genomic loci and, similar to snoRNA-derived miRNAs, requires Dicer but not Dgcr8 for their biogenesis [10,55]. Members of this subclass of ncRNAs are shRNA-miR-320, miR-484, miR-1980, and the isoleucine tRNA gene. They are all characterized by unique structural features, such as hairpin structures flanking the classic pre-miRNA hairpin that cannot be recognized by Dgcr8. The complex and unique hairpin structure is localized at the 3´ end and not at the canonical 5´end of these shRNA-derived miRNAs [10] (Figure 1). Additional data on shRNA-derived miRNA clusters are emerging, suggesting that a novel biogenesis might reflect other unperceived pivotal regulatory functions.

tRNA-derived miRNAs are produced during the maturation of tRNAs, but they share functions similar to those of miRNAs, such as the silencing of RNA transcripts. First data reporting the existence of tRNA-derived miRNAs emerged from HIV infected cells showing an increased number of tRNAs of approximatively 20 nt in length deriving from the HIV-transcriptional promoter tRNA-Lys3 [58,60]. These tRNAs showed the ability to bind Ago2 and to act as Dicer substrate, following a duplex tRNA formation [58]. Like snoRNA-derived miRNAs, tRNA-derived miRNAs are 18–22 nt in length and derive from both the 5´and 3´ends of mature tRNAs (Figure 1). Interestingly, Dicer knockout studies indicate that tRNA-derived miRNAs are more sensitive to Dicer deficiency than canonical miRNAs [58,60]. However, their binding to Ago proteins seems to be less stable than that of canonical miRNAs, probably due to their sensitivity to post-transcriptional modifications [58,60]. Their function is still unknown, but few data suggest that tRNA-derived miRNAs might support miRNA biogenesis and cell proliferation [58,60,61]

#### 4.2.2. MiRtrons

The miRtrons are the first class of non-canonical RNAs discovered in *Drosophila melanogaster* in 2007 [62]. MiRtrons originate from the introns processed by the spliceosomes during miRNA hairpin production. They can be processed by Dicer but not by the nuclear Drosha/Dgrc8 complex. Indeed, compared to pre-miRNA structures, miRtron pre-miRNA-like precursors show a shorter hairpin structure that cannot be processed by Drosha or Dgrc8. MiRtron pre-miRNA structures are instead debranched from the DBR1 enzyme [62]. Like miRNAs, miRtron precursors require the Exportin-5-bound stage to be transported in the cytoplasm and cleaved by Dicer [62] (Figure 1). Recently, high throughput sequencing and structural analysis data confirmed that miRtrons differ from canonical miRNAs in their CG content and hairpin length, conferring them more stability [63]. However, although similarities exist between pre-miRNA precursors and miRtrons from *D. melanogaster* and mammals, their ration is smaller in humans compared to flies and their function is unknown [62,63]. Moreover, 5′-tailed miRtrons seem to be more frequent in mammals compared to *D. melanogaster*, where 3′-tailed miRtrons represent the major fraction of all miRtron structures.

#### 4.2.3. IsomiRs

Deep-sequencing data indicate that for almost every miRNA sequence reported in the miRBase database, there is an equivalent one generated through a non-canonical process [53,56,64]. These unique human non-canonical miRNA isoforms are termed isomiRs and differs from their miRNA precursors in their length and sequence [53,56]. Although the mechanisms behind isomiR biogenesis are not fully understood, they might be generated by alternative Drosha/Dicer cleavages of miRNA precursors, insertions, deletions, substitutions, 5′ and/or 3′ cleavage variations, or 3′-end non-templated additions [53,64] (Figure 1). Additionally, isomiRs can be generated by posttranscriptional nucleotide additions within the sequences of miRNA precursors [64]. Cloonan et al. proposed a machine interpretable nomenclature based on the sequence of the isomiRs and the relative position of aforementioned modifications in the pre-miRNA hairpins to distinguish isomiRs from canonical miRNAs (see ref. [53] for more details). IsomiRs can share the same target of their miRNA precursors to regulate common [53] or different biological pathways [64,65]. Moreover, they can increase the mRNA-target specificity of canonical miRNAs [53]. However, because functional data are still scarce, most of the isomiRs discovered by sequencing still need to be included in the giant “junk RNA” group until their functions are demonstrated.

#### 4.2.4. Dicer-Independent miRNAs and Simtrons

Data reported so far indicate that Dicer is indispensable for the biogenesis of both canonical and non-canonical miRNAs. In contrast, Drosha is indispensable only for canonical miRNAs, whereas it is dispensable for the biogenesis of non-canonical miRNAs and miRNA-derived structures [10,39,66,67]. However, exceptions have been documented so far for certain miRNAs [66,68]. The first reported miRNA produced in a Dicer-independent fashion was miR-451. Indeed, plenty of data demonstrate that miR-451 biogenesis is independent from Dicer, an exception that, together with miR-451 function, is conserved in humans and mice [10,39,55,68]. Dicer-mediated cleavage of miR-451 precursor is hampered by a Drosha/Dgcr8 shortened cleavage of the pre-miR-451, leading to a too short (~18 nt) substrate for Dicer cleavage [69]. The pre-miR-451 is, therefore, cleaved by unknown enzymes at its 3′ end and loaded in the AGO2-RISC complex, or directly loaded in an alternative, AGO1-enriched, silencing complex [39] (Figure 1). However, recent data indicate that miR-451 is more enriched in the canonical AGO2-enriched RISC complex compared to those comprising alternative AGO proteins [39] and requires the activity of EIF1A, another RISC component, for its processing by AGO2 [70] (Figure 1). Hence, these studies not only indicated the existence of a Dicer-independent miRNA, but revealed how other components of the RISC complex, such as EIF1A, can augment AGO2-dependent unconventional miRNA biogenesis and function.

Although Dicer is indispensable for miRtron biogenesis, recent findings identified two exceptional human miRtron structures, named miR-1228 and miR-1225, which are produced in the absence of the canonical miRNA processing axis [71]. These non-canonical miRtron-like molecules are defined as splicing-independent miRtron-like miRNAs or “simtrons” and require only Drosha for their biogenesis [71]. Indeed, knockout of Ago2, Dgcr8, and Dicer does not affect simtrons biogenesis, whereas deletion of Drosha affects their maturation [71]. Simtrons and miRNAs share a similar function, interacting with Ago2 in the RISC complex for the silencing of their target genes [71]. Although their function is still not clarified, these miRNA-like molecules might share similar targets with canonical miRNAs.

Together with miR-451, recent findings demonstrate that another functional miRNA precursor, named pre-miR-2137, can be processed in a Dicer-independent fashion. Indeed, Hamin and colleagues uncovered a novel role for the inositol-requiring enzyme-1 (IRE1) RNase in the biogenesis of pre-miR-2137 [72] (Figure 1). IRE1 is an endoribonuclease activated under endoplasmic reticulum stress upon unfolded protein accumulation. Several works already demonstrated the involvement of IRE1 in miRNA regulation. Indeed, IRE1 regulates the expression of miR-316 and miR-148a by regulating the transcription of miRNA-related transcription factors [73]. The work from Hamid and colleagues demonstrates that IRE1 can directly control the maturation of an miRNA in a Dicer-independent fashion [72]. Indeed, inhibition of IRE1 reduces the expression of miR-2137, whereas the use of a recombinant IRE1 and an IRE1-wildtype/Dicer-deficient mouse model promotes pre-miR-2137 cleavage in an XBP1- and Dicer-independent manner [72] (Figure 1). Hence, this unprecedented role reported for IRE1 in modulating miRNA maturation suggests a more intricate mechanism by which miRNAs are produced and can modulate cell functions.

## 5. lncRNAs

lncRNAs are RNA sequences usually longer than 200 nt and classified according to their biogenesis, structure, and mechanism of action. In general, lncRNA genes derive from intronic, enhancer (eRNA), promoter, antisense, sense, or bidirectional genes [74]. Those lncRNAs transcribed from separate or intergenic genes are classified as long intergenic lncRNAs (lincRNAs) [74]. Based on their structure, lncRNAs can be further classified into linear lncRNAs, which include lincRNAs and eRNAs, and circular RNAs (circRNAs) [74]. Since the biogenesis of circRNAs significantly differs from that of linear lncRNAs, they are considered as non-canonical lncRNAs, whereas linear lncRNAs are classified as conventional—canonical—lncRNAs [38]. However, the non-conventional classification of all lncRNAs and circRNAs partly depends on their different, non-conventional, functions.

### 5.1. Canonical Biogenesis of lncRNAs

Like conventional mRNA transcripts, canonical lncRNAs are transcribed by RNA Pol II, and, similarly to mRNA transcripts, possess an m^7^G cap at the 5′ end and a polyA tail at the 3′ end of their sequence [74,75]. However, they differ from mRNAs for their transcription, export and turnover, all steps reflecting their versatile mechanism of action and cellular function [75]. Most lncRNAs exert their function in the nucleus [74]. Accordingly, after transcription, lncRNA transcripts are processed through alternative mechanisms compared to mRNAs and miRNAs [75,76]. This is partly reflected by their fine-tuned regulation upon transcription, which occurs in response to selective stimuli or during specific cell processes [75,76]. Indeed, the expression of lncRNA genes is usually repressed by histone modifications at their promoters, released upon selective cell signalling. In detail, the transcription of lncRNA depends on the phosphorylation state of the RNA Pol II carboxy-terminal domain, and on the trimethylation state of the lncRNA gene, fundamental for RNA Pol II activity [77]. Phosphorylation-dysregulated RNA Pol II cannot bind to the SPT6 elongation factor, thus producing numerous unstable lncRNA transcripts that accumulate on chromatin and are rapidly degraded by the RNA exosome [77]. Changes in the trimethylation state of H3 from lncRNA genes promote the proper RNA Pol II phosphorylation and binding to SPT6, which in turn promotes the accumulation of R-loop enriched lncRNA transcripts with a more stable structure [75,77]. Additional factors leading to lncRNA accumulation in the nucleus are the reduced splicing efficiency and the alternative polyadenylation system [75,77]. Indeed, due to a longer distance between the splice sites, lncRNA splicing is less efficient than that of mRNAs [74]. Moreover, both long and short tailored lncRNA isoforms can be produced by ribonuclease P-mediated cleavage and capping of their ends, leading to less stable structures. Hence, lncRNA transcripts are retained in the nucleus to avoid their degradation [74,77]. lncRNAs showing stable structures can be further processed and, therefore, transported in the cytoplasm through mechanisms similar to that of mRNAs. Cytoplasmic lncRNAs are shortened and associated with RNA-binding proteins through certain *cis*-elements localized primarily at their 5′ ends [78].

### 5.2. Non-Canonical Biogenesis of lncRNAs and circRNAs

Recent findings indicate that unique subnuclear structures, named “paraspeckles”, are also involved in lncRNA biogenesis [76,79]. Paraspeckle assembly transcript 1 (NEAT1) lncRNAs are produced through both canonical and non-canonical mechanisms that rely on paraspeckle subnuclear structures [79]. NEAT1 lncRNAs are exceptionally abundant mammalian lncRNAs and comprise two isoforms, named NEAT1_1 and NEAT1_2. They are transcribed by the RNA Pol II from the same promoter gene. Both lncRNAs lack introns and, therefore, are not produced through the canonical post-transcriptional splicing process. Indeed, they are processed at their 3′ ends to produce the canonical NEAT1_1 and the non-canonical NEAT1_2 transcript. The non-canonical transcript is characterized by a triple-helix structure that confers more stability to this isoform [79]. Moreover, electron microscopy data reveal that both structures are retained in the nucleus through the binding of the paraspeckle complex with their paraspeckle affinity regions located at their 5′ ends [79]. Although the function of paraspeckle lncRNAs is still uncertain, they might regulate cell response under certain stress conditions [79].

circRNAs were initially considered as a by-product of RNA splicing, generated through the junction between the upstream exon and downstream exon site. The typical “back-splicing” structure of circRNAs originates from the non-canonical splicing and covalently closed loop of pre-mRNAs. The circular structure protects circRNAs from RNase R degradation. circRNAs are generally divided into four groups, such as the exonic circRNAs, which comprise almost 80% of all identified circRNAs, the intronic circRNAs, the tRNA intronic circRNAs, and the exon-intron circRNAs. Although the biogenesis of circRNAs is not entirely known, three mechanisms have been widely proposed to generate circular RNAs from their linear precursors, such as the lariat-driven, the intron pairing-driven, and the RBP-mediated circularization mechanisms [80]. The intron pairing-driven circularization is the biological mechanism that generates the exonic circRNAs, which derive from an alternative circularization by complementary pairing of both sides of the intron during pre-mRNA processing [80].

## 6. Non-Canonical Function of ncRNAs in CVD

### 6.1. IsomiRs and miRNA Epitranscriptomes

The analysis of RNA-bound and polysome-associated miRNAs and isomiRs in human tissues reported by Cloonan and colleagues demonstrates that isomiRs are expressed at a biological level similar to that of miRNAs [53]. Since they are associated with the translational machinery, isomiRs are likely to be functional, exactly like miRNAs [53,56,64]. Using biotin-labelled synthetic miRNAs or isomiRs to pull-down endogenous mRNA targets, Cloonan et al. demonstrated that miRNAs and their isomiRs, such as miR-10a/b and isomiR-10a/b, act cooperatively to drive similar processes, such as cell cycle and apoptosis. Moreover, isomiRs might increase the effects of their respective miRNAs [53,64] (Figure 2a).

Recent findings indicate that miRNAs and isomiRs can regulate CVD by playing synergistic or antagonistic roles. Indeed, van der Kwast and colleagues demonstrated that miR-411, encoded from a large miRNA cluster located on the long arm of human chromosome 14, has a 5′-isomiR, named ISO-miR-411, which derives from a single nucleotide shift in Drosha’s cleavage of the pri-miR-411 [81]. The authors elegantly demonstrated the importance of the miRNA cluster located in the 14q32 chromosomic arm as one of the most regulated loci during ischemia, which plays a fundamental role in both the initiation and development of most CVDs. ISO-miR-411 is 5-fold more abundant than its respective miR-411 in the human vasculature of patients with peripheral artery disease (PAD) [81]. Moreover, compared to miR-411, it affects ischemia/neovascularization by targeting mRNA transcripts different from those targeted by miR-411 [81] (Figure 2a). Of note, this is the first work demonstrating a novel isomiR-mediated active regulation in a cardiovascular setting.

Stimulation of neovascularization is a potential strategy to promote vascular repair and to restore blood flow in ischemic patients. Accordingly, current therapeutic strategies promote both angiogenesis and arteriogenesis, since both are crucial to improve neovascularization [82]. However, unlike arteriogenesis, ischemic-mediated hypoxia triggers angiogenesis to resolve local ischemia Most of the known miRNAs playing a role in vascular biology and neovascularization show relative isomiRs (Figure 2a,b). Among these, miR-21-5p [83], miR-10a-5p [84], and miR-126-5p and -3p [85] are all well-established modulators of vascular biogenesis, and are all expressed together with their respective 5′-isomiRs [84]. Since the isomiRs are usually more abundant than their respective miRNAs [84], they might promote the expression and function of their respective miRNAs during vascularization. In addition, several studies on miRNA and isomiR expression under hypoxic conditions support the cooperative role exerted by certain isomiRs with their respective miRNAs. For example, miR-222-3p is expressed with its 3′-isomiR to modulate vascular cell inflammation during ischemic conditions, whereas the 5′-isomiRs deriving from miR-441 play an opposite role compared to miR-441, since they inhibit the translation of the pro-angiogenic Angiopoietin-1 in patients with PAD [81].

Key and novel modulators of neovascularization are certain non-canonical, modified forms of miRNAs named miRNA epitranscriptomes (Figure 2a,b). miRNA epitranscriptomes are generated through a canonical biogenetic program followed by certain biochemical nucleotide modifications that confers them novel functional regulatory roles (Figure 1). Indeed, miRNA epitranscriptomes are characterized by three main modifications: the adenosine editing (A-to-I editing), the N6-adenosine methylation (m6A), and the 2′-O-methylation (2′OMe) [57,65]. Functional data exist mainly for the A-to-I editing and the m6A methylation. Although most identified miRNA epitranscriptomes are involved in cancer progression, recent findings indicate that the A-to-I editing and the m6A modifications confer to miRNA epitranscriptomes the ability to regulate cardiovascular homeostasis (Figure 2b). Indeed, deep sequencing of murine and human muscle tissues collected during post-ischemic neovascularization by van der Kwast and colleagues indicates that miR-487b, upregulated during chronic hypertension and aneurysm formation, is edited in its seed sequence during ischemia [86]. In particular, the 2′OMe promotes a novel, proangiogenic function of miR-487b, which show a unique targetome compared to its canonical form during ischemia [86] (Figure 2b). Concomitantly, additional studies indicate that m6A modifications are required for the appropriate processing of most miRNAs involved in vascular homeostasis. As an example, the work from Ma et al. demonstrated that pri-miR-126 undergoes an m6A methylation that is mediated by methyltransferase 14 (METTL14) and facilitates the maturation of miR-126 [87] (Figure 2b). Hence, dysregulation of miRNA modifications during vascular biogenesis can significantly promote ischemic CVD and impair neovascularization.

### 6.2. Small-Derived miRNAs and MiRtrons

Certain snoRNA-derived miRNAs have been reported to share similar targets with miRNAs and to regulate several biological processes, such as cell proliferation and inflammation [88]. As an example, sdRNA-93 and sno-miR-28 regulate cell proliferation by repressing the p53-stabilizing transcription initiation factor TFIID subunit 9b (TAF9B) [88,89,90]. MiR-320 is dysregulated in ischemic hearts, where its inhibition using anti-miR-320 reduces cardiac infarction size [91]. Moreover, it is highly expressed in the pancreatic islets of Langerhans, where it suppresses insulin resistance.

The role of tRNA-derived miRNAs is still unknown. Recent data suggest that those loaded in the RISC complex might regulate mRNA transcripts similarly to canonical miRNAs. Others suggest that their loading into the RISC complex may support miRNA biogenesis. As an example, tRNA CU1276 can suppress the expression of RPA1, an enzyme crucial for DNA damage repair in cancer cells [61]. In addition, as with other miRNAs, stress-induced tRNA-derived miRNA FZD3 inhibits cell proliferation by targeting members of the Wnt signalling pathway [60]. However, it is still not clear if this function is exerted through the RISC complex. So far, only 3′tRNA fragments have been identified in the RISC, but their function is still unknown [92].

As the first discovered class of non-canonical miRNAs, miRtrons attracted the interest of researchers. Discrepancies between miRtrons and miRNA structures, as well as differences between mammals and flies in miRNA-yielding intron splicing methods, suggest the existence of distinct evolutionary origins and mechanisms of action. High-throughput sequencing data indicate that mammalian miRtrons might be involved in gene silencing [63], but data are still too preliminary.

### 6.3. Oxidative and ER Stress-Related Non-Canonical miRNAs

Non-canonical post-transcriptional modifications of miRNAs can promote the targeting of novel, alternative mRNA targets. The 8-oxoguanine (o^8^G) modification induced by reactive oxygen species (ROS) in certain miRNAs is one of the recent mechanisms promoting the miRNA-mediated progression of CVD (Figure 2c). As an example, the non-canonical ROS-mediated introduction of o^8^G in position 7 of the seed sequence of miR-1 (o^8^G-miR-1) promotes miR-1 targeting of novel mRNAs and cardiac hypertrophy [93,94]. Together with o^8^G-miR-1, Seok and colleagues demonstrated that miR-1 undergoes another ROS-mediated modification within its seed sequence. This variant is named 7U-miR-1, which derives from a G:U modification of the nt 7 in the miR-1 seed and can drive cardiac hypertrophy by affecting the cardiomyocyte phenotype [93] (Figure 2c). Hence, ROS-mediated oxidation of selective positions within the miRNA seeds can modify miRNA target specificity and generate an epitranscriptomic mechanism to modulate pathological gene expression [93].

IRE1 is a known ER-stress sensor and promoter of adaptive stress responses caused by the accumulation of unfolded proteins in the ER. IRE1 activity can be enhanced also by the incorporation of saturated fatty acids (SFA) or free cholesterol within the ER [72,73]. Indeed, although the ER plays a central role in lipid metabolism, it is poor in cholesterol and SFA. However, free cholesterol and SFA can accumulate in the ER in metabolic disorders and during chronic atherosclerosis [72]. Accumulation of SFA within the ER membrane dysregulates macrophage IRE1 and promotes its unconventional role as regulator of miRNA biogenesis [72,95] (Figure 2d). Indeed, IRE1 activation caused by ER lipid accumulation promotes the Dicer-independent and IRE1-dependent maturation of miR-2137, which targets and downregulates the inositol polyphosphate phosphatase-like 1 (INPPL1) transcript, a lipid phosphatase that acts in the PI3K-PKB-mTOR axis [72] (Figure 2d). Hence, the non-canonical IRE1-miR-2137 axis inhibits INPPL1-mediated inhibition of the mTOR pathway to enhance macrophage growth and proliferation and to promote atherosclerosis [72].

### 6.4. Non-Canonical Function of Nuclear and Mitochondrial miRNAs

Most canonical miRNAs exert their pro-atherogenic function in the cytoplasm by interacting with their target transcript in the RISC. In addition, recent data indicate that certain miRNAs involved in CVD development and progression exert their function in the nucleus or in the mitochondria (Figure 2e,f). The first evidence of a nuclear miRNA was reported by Meister and colleagues, demonstrating that 20% of total miR-21 levels, which mainly reside in the cytoplasm, were instead accumulated in the nucleus [66]. Notably, miR-21 promotes atherosclerosis and CVD by increasing cell apoptosis and inflammation in ECs and macrophages, respectively. Several other studies indicate that other miRNAs, such as miR-206, miR-122, and miR-29b, are actively transported into the nucleus and co-localize with nuclear proteins [96] (Figure 2e). Sequencing data identified three main mechanisms responsible for miRNAs shuttling into the nucleus. A common factor among all three mechanisms is the presence of selective signal sequences within the sequences of certain miRNAs. As an example, miR-29b is a member of the miR-29 family, but differently from miR-29c and miR-29a, it contains an AGUGUU-motif at the 3′ end that is responsible for its nuclear localization [97] (Figure 2e). Accordingly, additional studies identified this motif in the sequence of selective miRNAs capable of translocating into the nucleus and exerting their role by binding to specific RNA-binding proteins. However, it is not clear whether nuclear localization sequences are indispensable for miRNA shuttling, since certain nuclear miRNAs, such as miR-21, seem to be devoid of specific motifs.

Emerging evidence indicates that Dicer and RISC components such as IPO8 and AGO2 are involved in the shuttling of miRNAs between the cytoplasm and the nucleus [55] (Figure 2e). AGO2 emerged as the main protein responsible for nuclear miRNA shuttling. Although data regarding the role of nuclear miRNAs are poor, recent findings indicate that nuclear miRNAs can bind with high affinity to introns and to the 3′- and 5′-UTRs of mRNAs in a seed-dependent manner [98] (Figure 2e). Most of the genes regulated by nuclear miRNAs code for enzymes that drive epigenetic processes. Moreover, the same epigenetic proteins can regulate nuclear miRNA functions. This sub-class of nuclear miRNAs, named epigenetic miRNAs (epi-miRNAs), can interact, directly or indirectly, with epigenetic gene transcripts and components, such as HDACs, TETs, and HMTs [94] (Figure 2e). A known epi-miRNAs is the nuclear miR-29b, which can modulate the expression of epigenetic enzymes to inhibit the methylation of genes promoting cell proliferation and inflammation [94]. In addition, other nuclear miRNAs, such as miR-9, miR-24-1, miR-155, miR-212, miR-133a, let-7c, and miR-21-3p, promote histone acetylation and deacetylation of genes involved in cardiac hypertrophy and heart failure. As an example, Yan and colleagues demonstrated that nuclear miR-21-3p directly targets *Hdac8*, a member of the Class I histone deacetylases, to regulate the cardiac hypertrophic response [99] (Figure 2e).

Canonical miRNAs can regulate atherosclerosis through a non-canonical interaction with nuclear proteins involved in cell inflammation and apoptosis. (Figure 2e). Indeed, we recently reported that miR-126-5p directly binds and forms a ternary complex with AGO2 and MEX3A through its 5′ end to shuttle from the cytoplasm to the nucleus [100]. Once in the nucleus, miR-126-5p dissociates from AGO2 and binds with its seed sequence to the pro-apoptotic caspase-3 protein in an aptamer-like fashion, preventing caspase-3 dimerization and limiting its apoptotic function, a fundamental pro-atherosclerotic step [100] (Figure 2e). Hence, the direct interaction with and inhibition of proteins by nuclear miRNAs reveal novel, non-canonical, mechanisms by which miRNAs can modulate CVD.

Another non-canonical function of miRNAs is associated with their non-canonical localization in the mitochondria (Figure 2f). Even if it is not clear how they translocate into the mitochondria, it has been hypothesized that certain nuclear-encoded miRNAs, such as miR-143 and miR-146a, which play antagonistic roles against cardiac dysfunction [101] and atherosclerosis [102], might require AGO2 for their shuttling into the mitochondria [103] (Figure 2f). As an example, nuclear miR-146a translocates into the mitochondria and regulates mitochondrial function and cardiomyocyte apoptosis, a function that is decreased during ischemic reperfusion (I/R) [101]. In addition, recent findings identified miRNAs directly coded by the mitochondria genome (Figure 2f). These miRNAs are named mitomiRs and originate from the mitochondria-derived mRNA transcripts. They act as post-transcriptional regulators inside the mitochondria [104]. Most mitomiRs are reported to play crucial roles in heart failure and CV complications [103,104] (Figure 2f). As an example, mitomiR miR-181c play a role in cardiac remodelling by affecting complex IV of the mitochondria respiratory chain, promoting ROS production and cardiomyocyte apoptosis [103]. Additional mitomiRs, such as miR696, miR-146a, miR-30, and miR-92a, promote heart failure and CVD in humans by affecting—directly or indirectly—mitochondria genes, such as cytochrome-b, ATP synthase subunits, and genes involved in oxidative stress [105].

### 6.5. Antagonistic Functions of miRNAs and lncRNAs

Although miRNA and lncRNA biogenesis and function are distinct and separated, they can influence their reciprocal biogenesis and function. This novel level of gene regulation is classified as “non-canonical”. Indeed, an miRNA can influence two distinct biological processes by targeting an mRNA and lncRNA. Moreover, an lncRNA can promote the activation of a certain pathways (acting as epigenetic regulator, guide or enhancer), but can inhibit the same pathway by acting as sponge or decoy of an miRNA [106]. Therefore, we could divide the lncRNA-miRNA-mRNA axis into two classes, such as the canonical and non-canonical ones. The canonical axis can comprise all lncRNAs acting as competitive endogenous RNAs (ceRNAs), sponges, or miRNAs decoys, since the most prominent body of data reported on lncRNA:miRNA interaction regards cytoplasmic lncRNAs preventing miRNAs from binding with their mRNA targets [74]. The canonical sponge interaction occurs between the 3′ ends of the lncRNAs and the miRNA sites responsible for their interaction with AGO2. Wang and colleagues firstly described this interaction, identifying the lncRNA cardiac hypertrophy related factor (CHRF) able to sponge miR-489 to reduce a hypertrophic response in neonatal cardiomyocytes [107]. lncRNA competition with miRNAs to bind the same mRNA target can also be considered a canonical axis, which regulates myocardial remodelling after injury [74,75]. Since most lncRNAs play pivotal roles as epigenetic regulators in the nucleus, all lncRNAs that regulate miRNAs and general gene expression in the nucleus can be included in the canonical axis classification. As an example, lncRNA H19, a well-known regulator of atherosclerosis and skeletal muscle regeneration, can generate miR-675 through an alternative splicing of the first exon regulated by the RNA binding protein HuR, which is upregulated during cell stress [108]. In addition, miR-221/222 can be co-transcribed with Ang-362, an lncRNA important for angiotensin II-mediated cellular smooth muscle cell proliferation [109].

In contrast to lncRNAs acting as miRNA sponges or decoys, data regarding CVD-related lncRNAs as targets of miRNAs are poor. However, the lncRNA-miRNA non-canonical axis is emerging as a novel mechanism in CVD. Indeed, recent findings proved the existence of lncRNAs, highly conserved between mice and humans, acting as miRNA decoys [44,106,110] (Figure 2g). Like mRNAs, miRNAs can trigger lncRNA decoys in the RISC through a perfect base-pair complementarity. LncRNA MALAT1, involved in myocardial infarction and atherosclerosis, is currently one of the most abundant and well-characterized lncRNAs [110]. Canonical MALAT1 promotes pro-fibrotic responses in heart after myocardial injury by acting as miR-145 sponge [110]. However, MALAT1 can be degraded by miR-9 through a non-conventional mechanism [110]. Indeed, Leucci and colleagues demonstrated that MALAT1, which localizes in the nucleus, contains two miRNA recognition elements to which miR-9 can directly bind in an AGO2-dependent manner [110] (Figure 2g). Interestingly, this binding occurs in the nucleus, highlighting the existence of novel miRNA functions as post-transcriptional regulators in the nucleus. We also identified a novel lncRNA, named lncWDR59, targeted by miR-103 in the RISC of EC to promote atherosclerosis [44]. Indeed, differently from all canonical lncRNAs, lncWDR59 localizes in the cytoplasm where it promotes EC proliferation by interacting with the Notch1-inhibitor Numb. Thus, miR-103 interacts with lncWDR59 and induces its transcript degradation in the RISC to inhibit EC regeneration and promote atherosclerosis [44] (Figure 2g). However, data so far are limited, and only few miRNA:lncRNA interactions have been validated in vivo [44,110].

### 6.6. circRNAs

circRNAs are attracting increased attention among researchers worldwide due to their potential modulation of several physiological processes, such as parental gene expression and protein-coding formation [8]. circRNAs mainly derive from the back splicing of exons. Usually packaged in exosomes, circRNAs have been reported to act as sponges of proteins and miRNAs [80,111] (Figure 2g). Indeed, circRNA binding of miRNAs occurs in exosomes, a mechanism now considered to be a biomarker in several pathologies [8,80,112]. As members of the lncRNA subclass, circRNAs show a broad spectrum of action, acting as transcriptional, translational, and epigenetic regulators of gene expression, regulating mRNA splicing, or interacting with proteins [80]. Like circulating miRNAs, circRNAs are frequently associated with selective RNA-binding proteins, such as AGO2 and lipoproteins, and released in the circulation encapsulated in microparticles [8,80] (Figure 2g). Hence, like miRNAs and lncRNAs, circRNAs can be used as biomarkers in several CVDs [8,80].

The first evidence of circRNAs acting as sponges of miRNAs is credited to Hansen and colleagues [80,111]. They demonstrated that circRNAs contain miRNA-responsive elements that allow them to sponge miRNAs and act as competing endogenous miRNAs (ceRNAs) to promote the transcription of miRNA target genes [80,111] (Figure 2g). Hansen and colleagues demonstrated that ciRS-7 contains more than 70 conserved miRNA target sites to target miR-7 in the RISC complex [111]. Additional works indicate that circRNAs can act as ceRNAs of miRNAs even in the absence of selective miRNA-binding sites. As an example, circCSNK1G3 and circHIPK3 promote cell proliferation by sponging miR-181b/d [112] and miR-7 [113], respectively.

Like nuclear miRNAs and lncRNAs, circRNAs show non-canonical roles as sponges of certain proteins (Figure 2g). Indeed, circRNAs such as circPABPN1, circ-Foxo3 and circYap contain specific binding sites for RBS2 to inhibit the biological activity of certain proteins. In detail, circYap and circ-Foxo3 can inhibit the initiation of translation and cell cycle progression by interacting with eIF4G and CDK2, respectively [80].

## 7. Conclusions

ncRNAs play pivotal roles in the onset and progression of cardiovascular and metabolic diseases. For this reason, there is a growing research effort to develop novel and accurate techniques to study the complex world of canonical and non-canonical ncRNAs. Validation of ncRNA functions in CVD would offer the chance to develop novel and selective ncRNA-based therapies against cardiovascular and metabolic disorders. However, we are still far from confirming the real existence of all the ncRNA sub-categories identified so far, and thus, are far from having confirmed their real ability to play a functional (physiological or pathological) role.

The exponential increase in high-throughput and sequencing data is consistently increasing the number and variety of new and existing ncRNAs. However, it has not been accompanied by a parallel—fast-enough—scientific confirmation of their biological relevance, facilitating an understanding of whether they can be classified as “functional” or “junk” RNAs. A perfect example come from isomiRs, expressed at a similar biological level of miRNAs. Paradoxically, nearly all miRNAs have the potential to be co-expressed together with their corresponding isomiRs. However, data on their functional role are still poor, leading to the conclusion that most sequencing-discovered isomiRs must still be included in the “junk RNA” group. So, what is the biological function of these highly expressed non-canonical ncRNAs? One of the most promising hypotheses is that non-canonical ncRNAs may potentiate the effect of canonical miRNAs in response to certain pathological conditions. Indeed, canonical and non-canonical ncRNAs share common transcript targets. Non-canonical ncRNAs can target different genes but are involved in the same pathways regulated by their respective canonical ncRNAs [36]. In this scenario, non-canonical RNAs may confer a more flexible and rapid response to drive specific responses to certain physiological and pathological conditions.

## Figures and Tables

**Figure 1 biomedicines-10-00445-f001:**
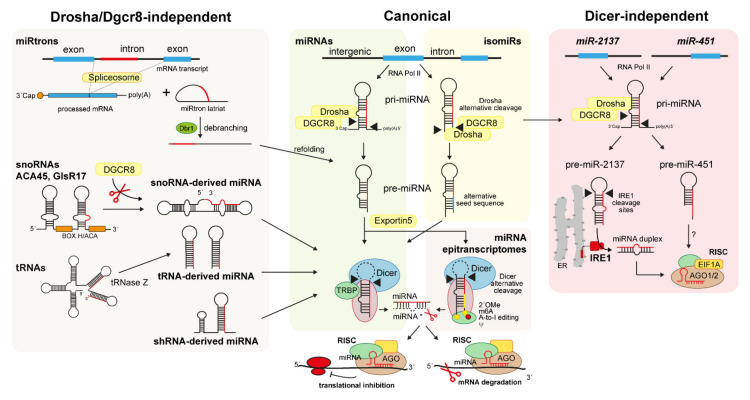
Canonical and non-canonical small RNA biogenesis. Canonical miRNAs and isomiRs are transcribed by RNA Pol II to generate a pri-miRNA precursor. Canonical or alternative Drosha/DGCR8 cleavage produces pre-miRNA or pre-isomiR precursors, which are exported from the nucleus to the cytoplasm through Exportin 5 to be further processed by the Dicer enzyme. Functional miRNA and isomiR guide strands are loaded in the AGO-enriched complex to regulate gene expression by translational inhibition or mRNA degradation though their canonical (red) or alternative seed sequences (red and blue). Alternative Dicer-mediated cleavage and second enzymatic processing of miRNA duplexes (e.g., 2´OMe, m6A, A-to-I editing) generate miRNA epitranscriptomes (alternative seed sequence in yellow), which can target novel mRNA targets. Drosha/DGCR8- independent biogenesis characterizes non-canonical biogenesis of small RNAs such as miRtrons, snoRNA-like miRNAs ACA45/GlsR17, tRNA-like miRNAs, and shRNA-like miRNAs. DGCR8 activity is required to generate miRNA precursors from snoRNA-like miRNAs. In addition, miR-451 and miR-2137 are produced through a Dicer-independent mechanism, due to a Drosha-mediated generation of too short pre-RNAs that cannot bind to Dicer. Indeed, miR-451 is processed from unknown enzymes and loaded into the AGO1-2 and EIF1A enriched RISC. Pre-miR-2137 is cleaved by ER-enriched IRE1 that generates a miRNA duplex with strands that can both be incorporated into the RISC.

**Figure 2 biomedicines-10-00445-f002:**
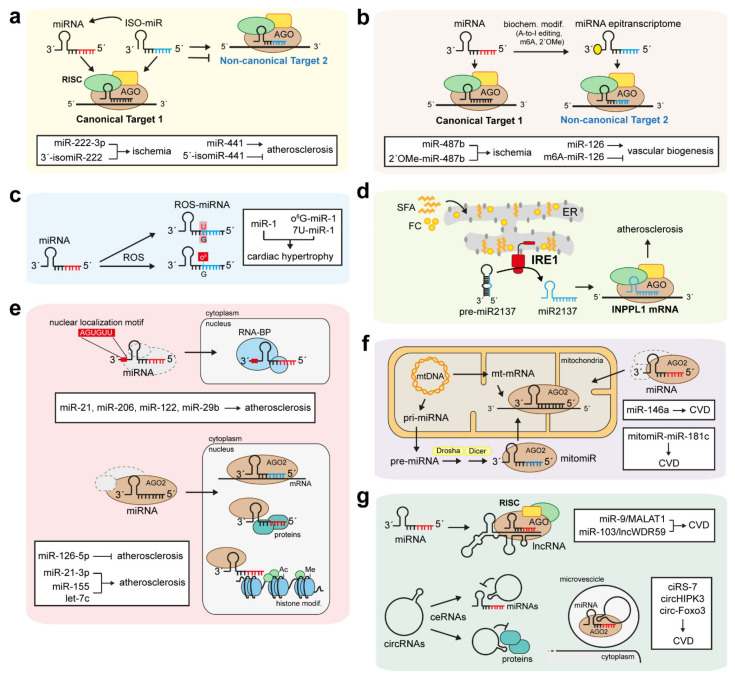
Non-canonical ncRNA function in CVD. In addition to their non-canonical biogenesis, miRNAs and lncRNAs show non-canonical mechanisms of action, which can be further influenced by their non-canonical biogenesis or post-transcriptional modifications. (**a**) IsomiRs produced through a non-canonical biogenesis can enhance miRNAs function by targeting the same miRNA targets or can play opposite roles by targeting other targets. (**b**,**c**) Post-transcriptional modifications of miRNAs promoted by ROS or biochemical factors enhance the role of miRNAs as promoters of CVD. (**d**) Non-canonical stress-induced lipid accumulation in the ER promotes IRE1-mediated miR2137 maturation, which promotes atherosclerosis by inhibiting the mTOR inhibitor INPPL1. (**e**) miRNAs can exert a non-canonical role in the nucleus, a shuttle mediated by AGO2. Nuclear miRNAs can interact and inhibit protein function, similarly to miR-126-5p that plays an anti-atherogenic role by inhibiting nuclear Caspase 3 activity. In addition, nuclear miRNAs can modulate the expression of genes by acting as epigenetic regulators (epi-miRNA), by targeting histone mRNA transcripts or directly influencing histone methylation (Me) and acetylation (Ac). (**f**) Non-canonical mitomiRs, transcribed by the mitochondrial DNA or shuttling in the mitochondria through AGO2, promote CVD by inducing mitochondrial mRNA (mt-mRNA) degradation and subsequent mitochondrial dysfunction. (**g**) lncRNAs can exert non-canonical roles by acting as miRNA targets, or by generating circular RNAs (circRNAs), which act as competing endogenous RNAs (ceRNAs) by interacting with miRNAs or proteins. Moreover, circRNAs can inhibit miRNA function by interacting with miRNAs in exosomes, in an AGO2-dependent manner. In red, the canonical seed sequence of miRNAs; in blue, the non-canonical seed or functional sequence. General miRNA and ncRNA sequences are in black.

## Data Availability

The data herein presented are available in this article.

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
