# Peer review of "A Non-Canonical Link between Non-Coding RNAs and Cardiovascular Diseases"

_biomedicines, 2022, doi:10.3390/biomedicines10020445_

Round 1

Reviewer 1 Report

As a whole, this is a reasonable/good review that covers roles of non-coding RNAs on cardiovascular function/disease.  The organization of the review is satisfactory.  However, there are several aspects that need to be improved/corrected.

  1. Writing must be carefully edited as there are many grammatical, punctuation, and  typographical errors.  For instance, the authors use "despite" as a conjunction.  "Despite" is a preposition.
  2. The following sentence is in Abstract.  "Despite the debate exist between the real existence of these non-canonical ncRNAs and their concrete biochemical function, making most of the dark genome being considered as “junk RNA”, in this review we will report those ncRNAs with a scientifically validated non-canonical biogenesis or mechanism of action in CVD."  It is not clear what the authors want to say here.  Since this is the key point of the review, please re-write using several sentences and describe fully the goal of this review.  "Despite" is used incorrectly here.
  3. Citations are insufficient.  For a review to be  useful, more original papers should be cited, especially for discussing CVD pathology.  In several places, WHO is credited for certain information.  Please give precise citations.  Lines 78-80. "Starting from the first report about the existence of an RNA of 15kb able to regulate the inactivation of the X chromosome, we arrived up to the definition that 80% of the genome encodes for ncRNAs, almost outclassing the role of protein-coding genes."  This sentence needs several pertinent citations.  Lines 183 and 185, references 75 and 76 are cited, but these references are not listed.
  4. Figures.  Parts of RNAs are colored (red, blue and black).  Please indicate the meaning of this in the legend.  

Author Response

As a whole, this is a reasonable/good review that covers roles of non-coding RNAs on cardiovascular function/disease.  The organization of the review is satisfactory.  However, there are several aspects that need to be improved/corrected.

  • Writing must be carefully edited as there are many grammatical, punctuation, and typographical errors.  For instance, the authors use "despite" as a conjunction.  "Despite" is a preposition.

We thank the Reviewer for the comment. The manuscript has been carefully revised and undergo through an extensive grammatical, punctuation, and typographical check and correction. All changes are indicated in the modified version of our Review.

  • The following sentence is in Abstract.  "Despite the debate exist between the real existence of these non-canonical ncRNAs and their concrete biochemical function, making most of the dark genome being considered as “junk RNA”, in this review we will report those ncRNAs with a scientifically validated non-canonical biogenesis or mechanism of action in CVD."  It is not clear what the authors want to say here.  Since this is the key point of the review, please re-write using several sentences and describe fully the goal of this review.  "Despite" is used incorrectly here.

We thank the reviewer for the constructive comment. Indeed, the abstract is essential to understand the goal of the Review. Accordingly, the sentence has been modified as follow “Moreover, recent studies reveal that canonical ncRNA sequences can influence the onset and evolution of CVD through novel “non-canonical” mechanisms. However, the debate exists between the real existence of these non-canonical ncRNAs and their concrete biochemical function, making most of the dark genome being considered as “junk RNA”. In this review, we will report the ncRNAs with a scientifically validated canonical and non-canonical biogenesis. Moreover, we will report canonical ncRNAs playing a role in CVD through non-canonical mechanisms of action.”

The manuscript has been revised to correct all grammar errors.

  • Citations are insufficient.  For a review to be useful, more original papers should be cited, especially for discussing CVD pathology. 

We thank the Reviewer for the comment. The goal of this Review was to summarize the data about ncRNAs with a scientifically validated role in CVD, focusing mainly on those showing a validated non-canonical biogenesis or a non-canonical mechanism of action. However, data are still poor, especially those clearly proving a role of non-canonical ncRNAs in CVD in vivo.

We agree that original papers regarding the CVD pathology are abundant but not sufficiently included in our Review. Accordingly, citations have been improved all over the manuscript. More citations have been included also in the Section 2, including those related to the global burden of CVD and CVD pathology.

  • In several places, WHO is credited for certain information.  Please give precise citations. 

Since direct references are not always available within the WHO page, the WHO web page links have been directly included in the text. This is the case for the “top 10 causes of death” reported from the WHO (https://www.who.int/news-room/fact-sheets/detail/the-top-10-causes-of-death),(https://www.who.int/data/gho/data/themes/mortality-and-global-health-estimates).

COVID-19 data were excluded but reported in the link https://www.who.int/data/stories/world-health-statistics-2021-a-visual-summary. All data have been included according to the World Helath Statistics 2021 (https://www.who.int/data/gho/publications/world-health-statistics).

In addition, we included the references for the Global Burden of CVD and Risk factors (Roth et al., 2020; Banerjee et al., 2020; Pearson-Stuttard et al., 2016).

  • Lines 78-80. "Starting from the first report about the existence of an RNA of 15kb able to regulate the inactivation of the X chromosome, we arrived up to the definition that 80% of the genome encodes for ncRNAs, almost outclassing the role of protein-coding genes."  This sentence needs several pertinent citations. 

Citations have been included.

  • Lines 183 and 185, references 75 and 76 are cited, but these references are not listed.

Citations have been included.

  • Figures.  Parts of RNAs are colored (red, blue and black).  Please indicate the meaning of this in the legend.  

We thank the Reviewer for the comment. The colors used in Figure 1 refer to canonical (red) seed sequences, proper of canonical ncRNAs, and alternative seed sequences generated by alternative cleavages (blue) or through second enzymatic processes (yellow). Accordingly, the meaning has been indicated in the figure legend.

The colors in Figure 2 are used in a similar way: miRNAs with a canonical seed sequence/function are in red, whereas in blue there are the non-canonical seeds, which bind to Non-canonical Targets (also in blue). Modification outside of the seed sequence have been designed accordingly. General miRNAs have been designed in black. As an example, in Figure 2e both canonical and non-canonical miRNAs can enter in the nucleus, therefore the cytoplasmic miRNA linked to AGO2 has been colored in black. Accordingly, the color definition has been included in the figure legend.

Reviewer 2 Report

This is a very comprehensive and well written review article about a relevant field. I don't have any comments or concerns.

Author Response

We thank the reviewer for the positive comments and are pleased to know that the article has aroused interest. Indeed, our aim was to encourage readers to consider cardiovascular disease from a non-canonical perspective.

Reviewer 3 Report

This review aims to discuss the relationship between ncRNA and CV diseases. The topic is quite intriguing and of interest. The review is well written and only a minor issue needs to be solved.

MINOR ISSUES:

  • Section 2, lines 47-59: mechanism of CV disease, hemodynamic and EC biology should be better sustained by referenced contributions. Accordingly, 2-3 references should be added about these sentences.

Author Response

This review aims to discuss the relationship between ncRNA and CV diseases. The topic is quite intriguing and of interest. The review is well written and only a minor issue needs to be solved.

 MINOR ISSUES:

  • Section 2, lines 47-59: mechanism of CV disease, hemodynamic and EC biology should be better sustained by referenced contributions. Accordingly, 2-3 references should be added about these sentences.

We are pleased to know that the article has aroused interest and we thank the Reviewer for the comments. We have revised the entire manuscript and increased the number of references within all Sections. Accordingly, the section 2 has been modified and references included.